# Validity of Ultrasound Imaging Versus Magnetic Resonance Imaging for Measuring Anterior Thigh Muscle, Subcutaneous Fat, and Fascia Thickness

**DOI:** 10.3390/mps2030058

**Published:** 2019-07-10

**Authors:** Filippo Mechelli, Lars Arendt-Nielsen, Maria Stokes, Sandra Agyapong-Badu

**Affiliations:** 1Centre of Sensory Motor Interaction, Department of Health Science and Technology, School of Medicine, University of Aalborg, 9220 Aalborg, Denmark; 2PT, MSc, Private Practice, 61029 Urbino, Italy; 3School of Health Sciences, University of Southampton, Southampton SO17 1BJ, UK; 4Centre for Sport, Exercise and Osteoarthritis Research Versus Arthritis, Nottingham NG7 2UH, UK; 5School of Sport, Exercise and Rehabilitation Sciences, University of Birmingham, Edgbaston B15 2TT, UK

**Keywords:** fascia thickness, MRI, muscle thickness, rectus femoris, ultrasound imaging, subcutaneous adipose tissue thickness, validity, vastus intermedius

## Abstract

The aim of the present study was to determine the validity of ultrasound (US) imaging versus magnetic resonance imaging (MRI) for measuring anterior thigh muscle, subcutaneous adipose tissue (SAT), and fascia thickness. Twenty healthy, moderately active participants (aged 49.1 ± 9.74 (36–64) years), underwent imaging of the anterior thigh, using ultrasound and MRI modalities on the same day. Images were analyzed offline to assess the level of agreement between US and MRI measurements. Pearson’s correlation coefficient showed an excellent relationship between US imaging and MRI for measuring muscle (r = 0.99, *p* < 0.01), SAT (r = 0.99, *p* < 0.01), and non-contractile tissue (SAT combined with perimuscular fascia) thickness (r = 0.99, *p* < 0.01). Perimuscular fascia thickness measurement showed a poor correlation between modalities (r = 0.39, *p* < 0.01). Intra-class correlation coefficients (ICC3,1) also showed excellent correlation of the measurements with ICC = 0.99 for muscle thickness, SAT, and non-contractile tissue, but not for perimuscular fascia, which showed poor agreement ICC = 0.36. Bland and Altman plots demonstrated excellent agreement between US imaging and MRI measurements. Criterion validity was demonstrated for US imaging against MRI, for measuring thickness of muscle and SAT, but not perimuscular fascia alone on the anterior thigh. The US imaging technique is therefore applicable for research and clinical purposes for muscle and SAT.

## 1. Introduction

Osteoarthritis of the knee [1], as well as other conditions that affect the knee [2,3], are commonly associated with quadriceps muscle weakness and atrophy (wasting). Quadriceps atrophy also occurs early and rapidly during critical illness [4,5]. The accurate, objective assessment of atrophy is a potentially powerful tool for research and clinical applications if a method is valid [6,7,8,9], and the present paper addresses this topic. Ultrasound (US) imaging provides an accurate, safe, and noninvasive tool, applicable in field environments with successful application in research and clinical practice to evaluate soft tissue structures of the musculoskeletal system. The technique is relatively cheaper than other imaging techniques, such as computed tomography and magnetic resonance imaging (MRI). The latter represents the most appropriate standard currently available for testing the validity against other methods [10].

Ultrasound imaging is an operator-dependent procedure [11]. Accuracy and reliability of US imaging for measuring anterior thigh subcutaneous adipose tissue (SAT) thickness have been recently demonstrated by Müller et al. [12] and Störchle et al. [13].

Repeatability of using the technique to assess muscle morphology between investigators (inter-rater reliability) and test–retest reliability have been established for different muscles across the age spectrum: Trapezius [14], supraspinatus [15], abdominal muscles [16], lumbar multifidus [17], gluteal (medius and minimus), and vastus medialis muscles [18]. Recently the intra-rater and inter-rater reliability of measuring anterior thigh tissues in a healthy middle-aged cohort were reported [19].

Validity of muscle thickness measurement using US against MRI is reported for different muscles, including cervical multifidus [20], supraspinatus and infraspinatus [21], trapezius [22], abdominal muscles [23], anterior hip muscles [24], and vastus medialis muscle [25]. Interest in fascia has grown in recent years [26,27,28,29], so it is important to determine the robustness of techniques for measuring fascia, as well as muscle.

Ultrasound techniques have improved over time, showing muscle tissue with resolutions up to 0.1 mm [30], better than an image obtained by a high-field MRI of 3 Tesla that reaches a resolution of 0.2 × 0.2 × 1.0 mm [31].

The validity for measuring anterior thigh tissue thickness (both muscular and non-contractile) requires examination. The present study aimed to examine the validity of US imaging in measuring muscle and non-contractile tissue thickness of the anterior thigh versus MRI, in healthy middle-aged individuals.

## 2. Materials and Methods

### 2.1. Participants

Twenty (10 females, 10 males) healthy, moderately active adults [32], aged 49.1 ± 9.74 years (36–64), with height (m) 1.72 ± 0.06 (1.59–1.82), and body mass (kg) 72.26 ± 11.42 (47.8–98.2) were studied. Participants were excluded if they had: Diseases and conditions affecting muscles (structure or function), musculoskeletal injuries of the lower limb and pathologies including fractures, surgical procedures, cancer, or neurological disorders. Participants were advised to refrain from vigorous exercise within the 24 h before being studied. The local Ethics Committee approved the study (CESU 1/2015). All participants were provided with full details of the study and then gave their written informed consent. The study was undertaken in adherence to the Declaration of Helsinki [33]. Participants’ rights were protected.

### 2.2. Procedure

All participants underwent imaging of both anterior thighs with US imaging and MRI on the same afternoon.

### 2.3. US Imaging Acquisition

The US imaging procedures have been described in detail in a previous publication, so only a brief outline is given here [19]. A US scanner (MyLab25; Esaote, Genova, Italia) with a 7.5 MHz linear transducer (40 mm length) in B-mode acquired transverse images of the anterior thighs. With the participant relaxed in supine lying (Figure 1), the hip was in neutral and the knee was in full extension. Scans were performed at a site two thirds of the distance measured from the antero-superior iliac spine to the superior pole of patella [34], and the site was marked with a skin marking pen. For image acquisition, US gel was placed over the marked site and the US transducer placed on the skin with minimal contact pressure to avoid distorting the tissues [12,35]. The same investigator (FM) took all the scans.

### 2.4. MRI Acquisition

Participants underwent a metal safety check prior to undergoing MRI scanning (Magnetom C! 0.35T, Siemens, Germany), which was performed using a body coil, with the participant in the same supine position and at the same level on the thighs used during US imaging (Figure 2). Vitamin E capsules were placed over the scanning sites on the thighs. T1-weighted images of both thighs were obtained using the following parameters: TR = 512 ms, TE = 15 ms, acquisition matrix = 256 × 256, slice thickness = 7 mm, FoV 180 × 180, pixel spacing 0.3515625\0.3515625.

### 2.5. Image Processing

US and MRI images were anonymized and analyzed offline by the same investigator (FM), using ImageJ software (https://imagej.nih.gov/ij/). Each image was measured twice and the mean of the two used in the analysis. SAT thickness was measured from the skin to the outside edge of the superficial fascial layer, while muscle thickness of the rectus femoris (RF) and vastus intermedius (VI) was measured between the inside edges of muscle borders to exclude the perimuscular fascia. The superficial fascia was measured between its outside edges, where it lay between the SAT and superior border of the RF, while the deep fascia lay between the RF and VI (Figure 3 and Figure 4).

### 2.6. Data Analysis

Data was analysed using SPSS 22 (SPSS Inc, Chicago, IL, USA) software package. The data were normally distributed on testing with the Shapiro–Wilk test. Descriptive statistics summarized the data as means and standard deviations. Pearson’s Correlation Coefficient (r) examined the correlation between the two imaging techniques. To assess the agreement between US imaging and MRI measurements, intra-class correlation coefficients (ICC_3,1_) were used. Bland and Altman analysis was used to assess the degree of agreement between the two imaging techniques, and detect bias and outliers. An ICC value of 0.9 or above was considered excellent and suitable for clinical measurements and diagnosis [36].

## 3. Results

### 3.1. Relative Measurements and Correlation Analysis

Muscle and non-contractile tissue measurements from MRI and US scans are shown in Table 1. Excellent correlation between US imaging and MRI measurements was demonstrated for muscle thickness (r = 0.99, *p* < 0.01), SAT (r = 0.99, *p* < 0.01), and non-contractile tissue (r = 0.99, *p* < 0.01). Perimuscular fascia thickness demonstrated a poor level of correlation of r = 0.39, *p* < 0.01 (Table 2).

### 3.2. Agreement between Modalities

The ICC analysis showed excellent agreement between the two modalities with ICC_3,1_ > 0.90 for all measurements except perimuscular fascia (ICC_3,1_ = 0.36), as reported in Table 3. Bland and Altman plots supported ICC data, demonstrating excellent agreement between US imaging and MRI measurements, with only one outlier and no bias (Figure 5, Figure 6 and Figure 7).

## 4. Discussion

The present study demonstrated the validity of US imaging compared to MRI for measuring thickness of the anterior thigh muscle and non-contractile tissues in a group of 20 healthy middle-aged individuals. Analyses showed excellent correlation and agreement between measurements of the two imaging modalities for measuring thickness of muscle, SAT, and non-contractile tissue, but perimuscular fascia measurements alone did not show good agreement.

The present findings compare favorably with previous validity studies, which compared US imaging and MRI measurements for other lower limb muscles, e.g., [25] reported good correlation (r = 0.87) between US imaging and MRI in linear measurements of vastus medialis distal fibres in a group of 12 healthy young adult males (aged 18–30 years). Another study also reported no significant difference (*p* > 0.05) in measurements of cross-sectional area (CSA) and volume of the quadriceps muscle between US imaging and MRI in a group of 10 healthy volunteers [37]. Mendis et al. [24] reported high agreement between US imaging and MRI CSA measurements of iliopsoas, sartorius, and rectus femoris muscles with ICC values ranging from 0.81 to 0.89 in a group of nine healthy individuals (mean age: 24.3 ± 3.5 years). Similar excellent agreement was reported (ICC 0.78–0.95) for thickness of the lateral abdominal muscles (transversus abdominis and internal oblique muscles), using US imaging and MRI in healthy male athletes with a mean age of 21 (SD 2.1) years [23].

Regarding the poor correlation and agreement results between US and MRI thickness measurements for fascia, reliability of measuring the fascia on the anterior thigh was also found to be poor [19]. The thickness of the fascia is only 2.6 mm, compared to the anterior thigh muscle, 28.1 mm and SAT 10.5 mm (Table 1), which would have greater room for error, potentially affecting validity and reliability.

Another potential issue regarding the poor correlation and agreement results for fascia could be due to the limit of the low field the MRI machine operates (0.35T). On the other hand, the resolution of the US image of the fascia was good even when the frequency was set at 7.5 MHz to improve ultrasound penetration and visualize deeper structures (at the expense of the image resolution of the superficial tissues, where a higher frequency would have been optimal).

The implications of these findings need to be considered when studying fascia. It may be more clinically relevant to evaluate the integrity and continuity of the fascia, due to its role of transmitting mechanical tension resulting from muscle activity [26,27,29], rather than measuring its thickness. However, differences in thickness of fascia on US images have been documented in patients with lower back pain, who demonstrated thinner abdominal wall muscles and thicker fascia than healthy controls [28]. The reliability of thickness measurements of muscle was reported as excellent in that study, but reliability of fascia measurements was not reported.

Compared to MRI, US imaging is safer, non-invasive, less expensive, and relatively faster with portable scanners for use in clinical and field environments. The MRI procedure may be claustrophobic for some individuals. Recently, there has been growing interest in the use of US imaging to assess muscle thickness at the bedside, particularly in intensive care settings. Critically ill patients may experience early-stage skeletal muscle wasting and monitoring such muscle mass loss using US imaging may help predict clinical musculoskeletal outcomes. The US technique could be a valuable, accurate, and accessible tool for the clinician to improve nutritional delivery to stop or attenuate loss of lean body mass [7,9]. A potential use of the technique could be for monitoring the effects of nutrition (different diet protocols or weight loss/gain programs) to ensure maintenance or gain in lean mass and a reduction of SAT [38,39,40]. The use of US imaging in ageing studies has shown the sensitivity to changes in muscle and SAT thickness with older age [41]. Takai [42] reported the potential of using US to predict fat-free mass in older people. There is evidence to support the use of US imaging in addition to DXA (Dual-energy X-ray absorptiometry) to provide measurements of quadriceps and individual muscle groups for earlier detection of sarcopenia [43], which is the joint loss of strength and impaired physical performance that is common in older people [44]. In clinical settings, US imaging of the anterior thigh could provide information on muscle wasting in patients with knee osteoarthritis [1] or other painful knee conditions [2,3]. Ultrasound muscle measurement may provide evidence on atrophy in astronauts due to prolonged microgravity and used to monitor effects of exercise interventions/training, intended to increase quadriceps muscle mass to provide accurate documentation of gain in lean mass and/or a reduction of SAT. This could be a powerful tool for studying the type and dose of exercise, which optimizes muscle hypertrophy, particularly for research on athletes and in sports settings.

Limitations of the present study include the participant population, which was limited to healthy middle-aged adults. It is unclear whether the correlations between US imaging and MRI would be as high in older adults, where the quality of images acquired may not be as clear as in a younger group.

## 5. Conclusions

The present findings provided further evidence of the clinimetric properties of US imaging. Specifically, US imaging has been demonstrated as a valid method for measuring anterior thigh muscular and non-contractile (combined SAT and perimuscular fascia) tissue thickness in healthy middle-aged adults, supporting its research, sports, and clinical applications. However, measurement of the thickness of fascia alone was not valid and needs to be considered in studies focusing on fascia.

## Figures and Tables

**Figure 1 mps-02-00058-f001:**
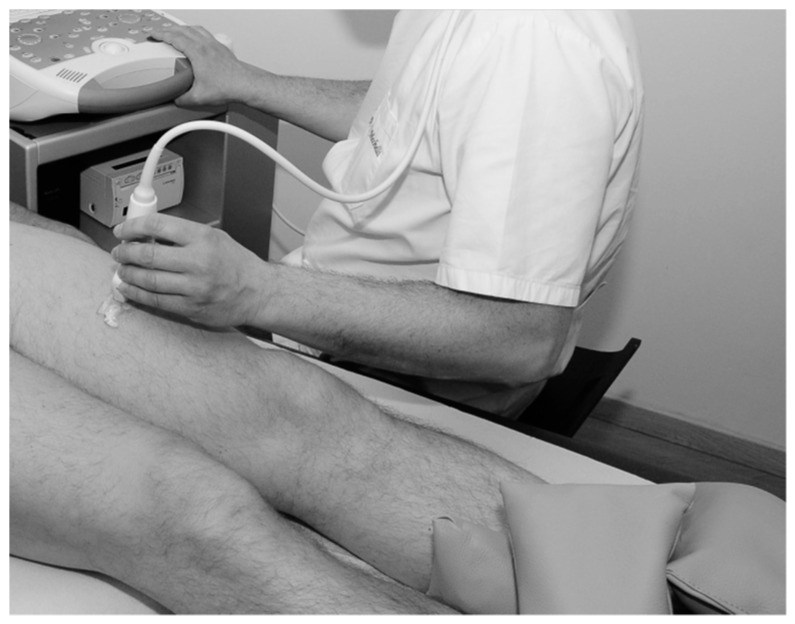
Ultrasound (US) scanning procedure with a participant lying supine and a US transducer placed on the anterior thigh.

**Figure 2 mps-02-00058-f002:**
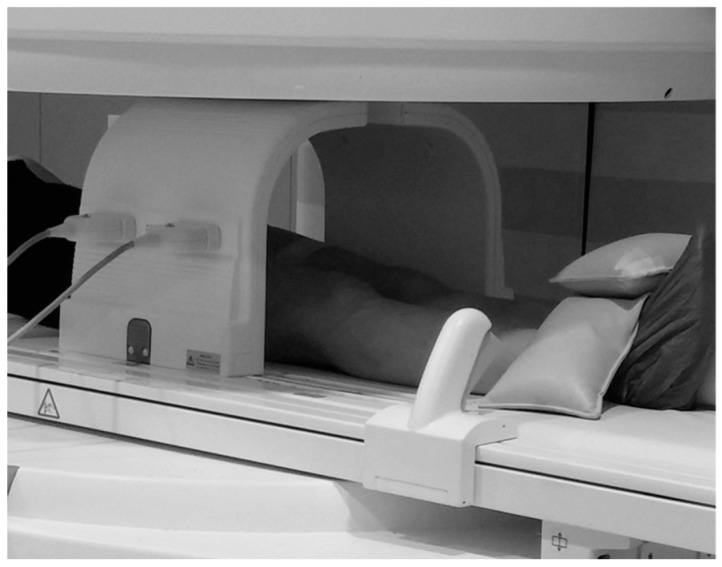
Magnetic resonance imaging (MRI) scanning procedure, with a participant lying supine, anterior mid-thigh under MRI body coil, and pillows at the ankle to maintain the hip in neutral.

**Figure 3 mps-02-00058-f003:**
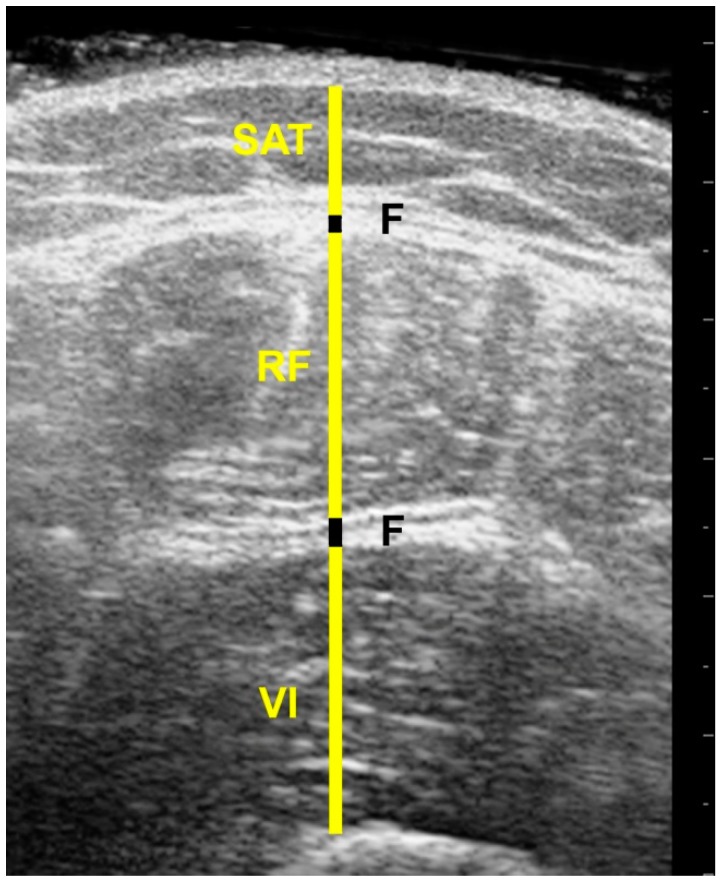
Ultrasound image of the anterior thigh; SAT= subcutaneous adipose tissue, F = Fascia, RF = Rectus Femoris muscle, VI = Vastus Intermedius muscle.

**Figure 4 mps-02-00058-f004:**
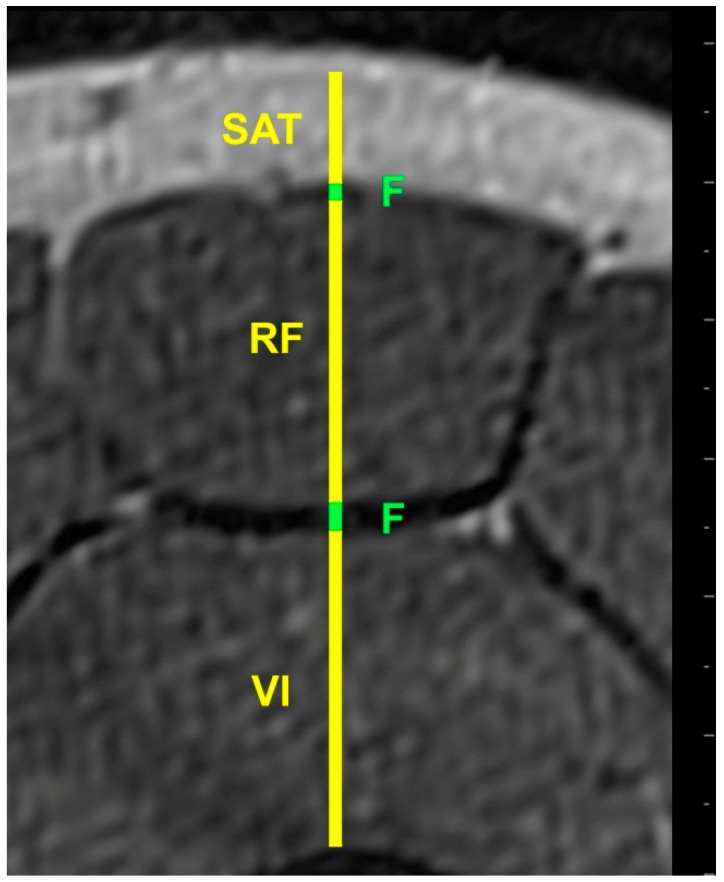
MRI image of the anterior thigh of the same participant; SAT = subcutaneous adipose tissue, F = Fascia, RF = Rectus Femoris muscle, VI = Vastus Intermedius muscle.

**Figure 5 mps-02-00058-f005:**
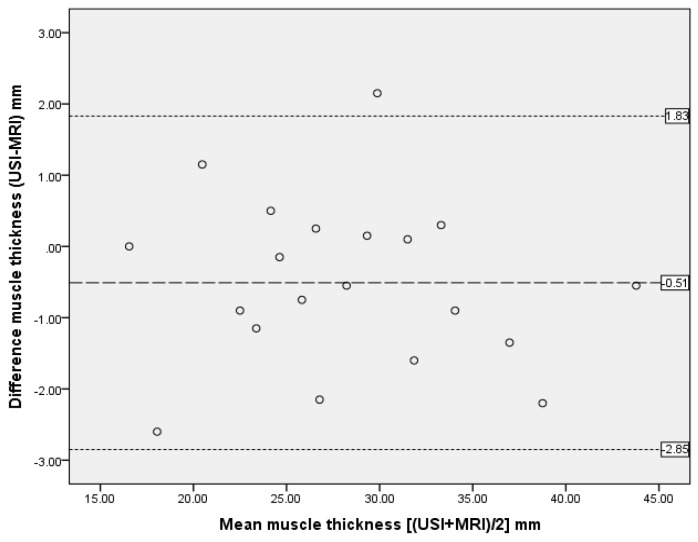
Bland and Altman plot showing difference between measurements of muscle thickness from MRI and US scan. The dashed line represents the mean difference; dotted lines are 95% upper and lower limits of agreement, representing two standard deviations.

**Figure 6 mps-02-00058-f006:**
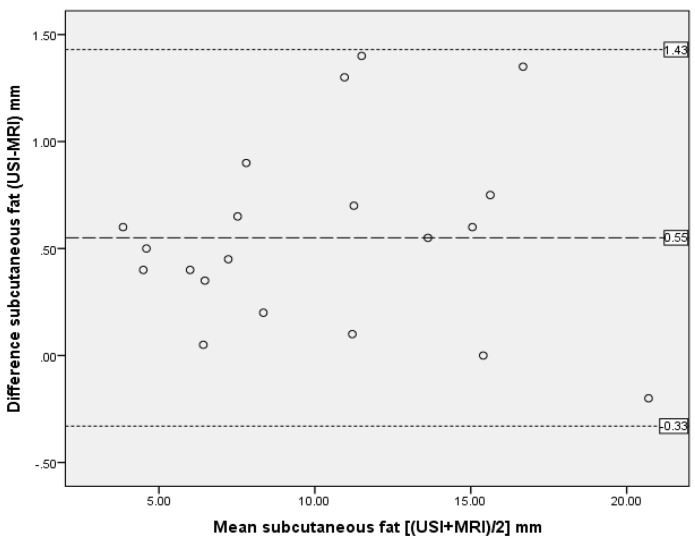
Bland and Altman plot showing difference between measurements of SAT thickness from MRI and US scan. The dashed line represents the mean difference; dotted lines are 95% upper and lower limits of agreement, representing two standard deviations.

**Figure 7 mps-02-00058-f007:**
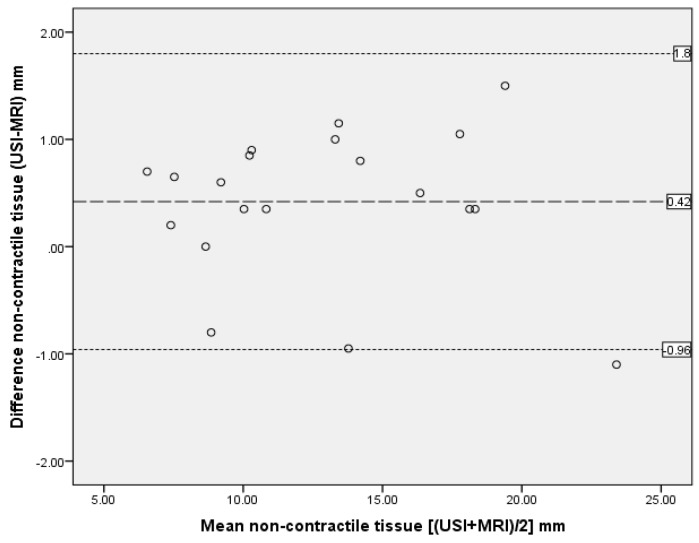
Bland and Altman plot showing difference between measurements of non-contractile tissue from MRI and US scan. The dashed line represents the mean difference; dotted lines are 95% upper and lower limits of agreement, representing two standard deviations.

**Table 1 mps-02-00058-t001:** Thickness of different tissue measurements on MRI and US.

Participants (n = 20)	Muscle Thickness (mm)	Subcutaneous Adipose Tissue (mm)	Non-Contractile Tissue (mm)	Fascia Thickness (mm)
Mean ± SD				
MRI	28.6 ± 7.1	9.9 ± 4.7	12.7 ± 4.7	2.7 ± 0.3
Ultrasound	28.1 ± 6.9	10.5 ± 4.7	13.1 ± 4.7	2.6 ± 0.4

**Table 2 mps-02-00058-t002:** Correlation between MRI and US measurements.

Participants (n = 20)	r	*p*-Value
Muscle thickness (mm)	0.99	<0.01 *
Subcutaneous adipose tissue (mm)	0.99	<0.01 *
Non-contractile tissue (mm)	0.99	<0.01 *
Fascia thickness (mm)	0.39	0.08

* Significant (two-tailed) at 0.01 level; r = Pearson correlation coefficient.

**Table 3 mps-02-00058-t003:** Comparison of measurements between MRI and US scans by intra-class correlation coefficients (ICC) and Bland and Altman analysis.

Anterior Thigh Measurment n = 20	ICC3,1	95% CI	SEM	Bland Altman Analysis		
				Mean Differences (mm)	Standard Deviation of Differences (mm)	95% Limits of Agreement (mm) Mean ± 2SD
Muscle thickness	0.99	0.965–0.994	0.69	−0.51	1.17	−2.85 to 1.83
Subcutaneous adipose tissue	0.99	0.989–0.998	0.47	0.55	0.44	−0.33 to 1.43
Non-contractile tissue	0.99	0.973–0.996	0.47	0.42	0.69	−0.96 to 1.8
Fascia	0.36	−0.084 to 0.687	0.29	−0.13	0.42	−0.97 to 0.71

CI = Confidence interval; SEM = standard error of measurement.

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
