# Peer review of "Validity of Ultrasound Imaging Versus Magnetic Resonance Imaging for Measuring Anterior Thigh Muscle, Subcutaneous Fat, and Fascia Thickness"

_mps, 2019, doi:10.3390/mps2030058_

Round 1

Reviewer 1 Report

The methods for measurement of layer thickness of each tissue are not clear.

For B-mode image and MRI image, the mark of each thickness which are measured should be shown in fig.3 and fig.4 just like fig.2 in ref.19. 

For MRI data, spatial resolution or pixel length should be shown.

Also it is not clear how you extract the cross sectional plane in MRI correspondent to B-mode image. The procedure should be written in detail.

The discussion of the error of measured values for the thickness should be related the spatial resolution of B-mode image and MRI.

Author Response

Comments and Suggestions for Authors

The methods for measurement of layer thickness of each tissue are not clear.

For B-mode image and MRI image, the mark of each thickness which are measured should be shown in fig.3 and fig.4 just like fig.2 in ref.19. 

For MRI data, spatial resolution or pixel length should be shown.

Also it is not clear how you extract the cross sectional plane in MRI correspondent to B-mode image. The procedure should be written in detail.

The discussion of the error of measured values for the thickness should be related the spatial resolution of B-mode image and MRI.

RESPONSE:

Each thickness layer measured is now shown in Figures.

The text has been amended with the following information:

FoV 180*180, pixel spacing 0.3515625\0.3515625  line 97

Cross-sectional plane in MRI corresponding with B-mode image was found by placing vitamin E capsules over the pen marks for the US scanning sites on the thighs. line 94-95

Reviewer 2 Report

General comments:

The research question is of interest and worth to be studied, however there are major shortcomings and mistakes to be corrected, and additional information is necessary.

The authors cite relevant publications and embed their work in the already existing scientific surrounding, but some references are cited in a wrong context. Further, the used methods are not state of the art; however, the main research outcome still remains useful. The methods can be improved partially, and the remaining methodical limitations need to be specified.

The authors use the term "gold standard" for their MRI measurements – this term is not appropriate for what they are studying. The authors use this expression without reflexion what they mean by that. E.g. when studying the MRI image (Fig.4), their image does not allow to distinguish between the skin and the subcutaneous adipose tissue (SAT); the authors use the sloppy term subcutaneous fat – it is not pure fat, the SAT consists of lipids, water, and adipose cellular structures, and additionally fibrous structures  are embedded in the SAT (also termed "fasciae"). How can a method be referred to as the "gold standard" if important image information is not contained (like skin/SAT border, embedded fasciae). In this concern, the US image should be seen as the "gold standard" because there is no other method that can image the SAT and the skin showing such fine details. In particular, the authors use an old MRI machine with 0.35 T and therefore have to use a very thick slice of 7 mm – thus, all they can get are "average" data over quite large volume elements, whereas US provides information from very thin slices and has a much better resolution than the MR images (this becomes directly obvious by comparing Fig. 3 and 4).

The US image shown is also not of the best quality (e.g. the epidermis can usually also be clearly imaged in state-of-the-art US images). It seems that the authors measure the thicknesses from their US images using the implemented thickness measurement tool of their US system. They have missed the latest developments of accurate measurement of thicknesses with ultrasound, although they cite two of the papers where this has been described in detail (35, 37).

Specific comments:

Abstract

L23: Use: a "poor" correlation instead of "lower".

L28: Avoid using the term "gold standard" for the MRI measurements in a general way (see general comments). Describe the comparison with MRI in a neutral way, based on facts (data of the MRI imaging system, image resolution etc.) and not based on this sloppily used term.

Introduction:

L42: Use a technically correct and neutral description for the comparisons you made. At least for measurements of SF thickness, when performed correctly (which is not the case in this publication), US can be seen as the method with the highest accuracy; this should be discussed in detail using the relevant citations (in the Introduction and in the Discussion sections). The latest publications dealing with accuracy and reliability of US measurements for thickness measurements of SAT are missing, e.g. the recent papers published in the Br J Sports Med (Müller et al., 2017), or in Ultrasound in Med&Biol (Störchle et al., 2017).

43: It is misleading to call the method that is the most expensive one automatically the "gold standard" (although, this misuse of this term "gold standard" is widely spread).

The MRI measurements made, at least for SAT measurements and with the MRI system used, do not represent "the standard" for testing validity, in the opposite, US – with its much better image resolution – is the standard for the purpose of SAT thickness measurements (it has been explained in the literature why US, when applied in standardised way, results in a thickness measurement accuracy (about 0.1 mm at 18 MHz) that is currently not reached by any other method including MRI (due to the much larger pixel/voxel sizes in MR images).  For example: the authors used a slice thickness of 7 mm, this value is almost 2 orders of magnitude above the US image resolution.

46:  "reliability". Also, discuss the recent papers on reliability of SAT measurement with US in detail as this is of essential concern for the study presented here. Compare to Br J Sports Med (A: Müller et al., 2017) and to B: Ultrasound in Med&Biol (Störchle et al., 2017). 

You cite papers on reliability, but do not discuss the accuracy obtainable with MRI (which depends on the measurement parameters, the pixel size, and the tissue segmentation program used). The obtainable accuracy and reliability will later in the manuscript have to be discussed in the context of the particular MRI setting the authors have used.

50: avoid  the term "gold standard", or give detailed explanations why you mean that this is the "gold standard" (in terms of accuracy and reliability at given MRI machine settings). Just writing that MRI is "seen as the gold standard" (by other authors) is not a scientific argument. Don't follow unreflected statements of other authors without analysing them on a scientific basis. 

Materials and Methods:

2.3 US:

Specify the speed of sound you used for thickness measurements. Consider that speed of sound differs significantly from 1540 m/s (as used in most US machines) in fat (you cited ref. #37, but did not use the information contained in this publication concerning the analysis of accuracy; also see ref A).

80: The authors obviously did not apply any pressure: according to Fig. 4, the authors used the method for avoiding compression artefacts as described in ref. 37 (a thick gel layer above the skin), but they do not cite this methodical procedure aspect as described in the ref. #37 of their manuscript (and in ref. A, too).

The authors should describe in detail how they measured the thicknesses in the US images (just one measurement with the cursor? average of how many measurements? What sound speed was used in the machine? What thickness measurement error is to be expected because of the wrong speed used, etc.)

2.4 MRI:

Information on the field of view and on the pixel/voxel size and form is missing. Why did the authors not use the DIXON technique to show the skin separately? (maybe to be discussed in "Limitations").

Use precise terms: you measured SAT and skin thickness together – this is not identical to "subcutaneous fat (SF)". Use precise wording. Further, define clearly what you mean by "non-contractile tissue" when you use this term for the first time. Describe the core parameters of the image analysis software and what software setting was used in your study.  

93: wording again

Results:

The results are interesting (although the methods were not used in a state-of-the-art manner) and statistical analyses appear to be correct and useful. The authors should point out that mean SAT thickness was larger when measured with US (and in the discussion, the authors should explain why this was the case – because of the wrong speed of sound for fat; compare to ref # 37 and ref A). It seems that the authors did not read their cited literature thoroughly; otherwise, they would have figured this out themselves already.

Figures:

Figs. 3 and 4: The depth of the image is not indicated.

Fig. 7: define what is meant by non-contractile tissue (remark: "SF" is also "non-contractile" !)

Description on axes is hard to read (letters too small).

Each figure with the figure legend should be self-explaining.

Discussion:

141: gold st. again

145: Here, I recognise the definition of non-contractile tissue for the first time. This should occur earlier.

159: Fasciae are complex structures, sometimes consisting of several layers, their thicknesses can change from one mm to another and so on (e.g. described in detail in: C Stecco: Functional Atlas of the Human Fascial System, 2015, Elsevier)  – this can be detected by US but not by MRI, at least not with the MRI machine and setting used. The authors should point out that MRI (their machine and their setting, and also what can be obtained with state-of-the-art MRI machines), therefore, cannot reach the accuracy of US (provided the appropriate speed of sound is used).

170-191: The authors embed their results into other research findings in a suitable way and draw useful conclusions.

180: ".. to change in muscle and SF thickness with older age [35-39]."

Several of the cited papers do not deal with aging studies at all. What do the authors mean here? These citations are in a wrong context.

192: Limitations: The limitations of the authors' approach from the technical point of view are missing. It is also missing that skin and "SF" (this term should be replaced or clearly defined) are not separately treated (skin thickness depends substantially on the site used, and also on the person under study). It is also missing why the authors did not use a standardised measurement approach as was described for SAT measurements in the literature. It has to be noted that their thickness measurements start out from a speed of sound that is not matched with the given tissues – in this concern, the authors can argue that conventional machines are not able to choose the appropriate speed of sound and therefore their approach mirrors the usual medical B-mode US applications.   

Author Response

General comments:

The research question is of interest and worth to be studied, however there are major shortcomings and mistakes to be corrected, and additional information is necessary.

The authors cite relevant publications and embed their work in the already existing scientific surrounding, but some references are cited in a wrong context. Further, the used methods are not state of the art; however, the main research outcome still remains useful. The methods can be improved partially, and the remaining methodical limitations need to be specified.

The authors use the term "gold standard" for their MRI measurements – this term is not appropriate for what they are studying. The authors use this expression without reflexion what they mean by that. E.g. when studying the MRI image (Fig.4), their image does not allow to distinguish between the skin and the subcutaneous adipose tissue (SAT); the authors use the sloppy term subcutaneous fat – it is not pure fat, the SAT consists of lipids, water, and adipose cellular structures, and additionally fibrous structures  are embedded in the SAT (also termed "fasciae"). How can a method be referred to as the "gold standard" if important image information is not contained (like skin/SAT border, embedded fasciae). In this concern, the US image should be seen as the "gold standard" because there is no other method that can image the SAT and the skin showing such fine details. In particular, the authors use an old MRI machine with 0.35 T and therefore have to use a very thick slice of 7 mm – thus, all they can get are "average" data over quite large volume elements, whereas US provides information from very thin slices and has a much better resolution than the MR images (this becomes directly obvious by comparing Fig. 3 and 4).

The US image shown is also not of the best quality (e.g. the epidermis can usually also be clearly imaged in state-of-the-art US images). It seems that the authors measure the thicknesses from their US images using the implemented thickness measurement tool of their US system. They have missed the latest developments of accurate measurement of thicknesses with ultrasound, although they cite two of the papers where this has been described in detail (35, 37).

Response:

The authors recognize that use of the term "gold standard"  for MRI imaging measurements is not appropriate, despite it being used widely in the literature for this purpose, and has been removed throughout the manuscript. In the introduction, MRI has been described as the most appropriate standard currently available for comparison, in order to justify its use in validity studies.

Subcutaneous fat (SF), is not solely fat and  has been replaced throughout the manuscript with the term subcutaneous adipose tissue (SAT).

It is indeed true that US imaging provides a much better resolution than MR images, especially for superficial tissues using an very high frequency transducer up to 18-22 MHz. However, the present study used  a lower frequency, as the purpose was also to measure  deeper structures down as far as the surface of the femur, so that  measuring thickness of the various tissues layers throughout the thickness of the anterior thigh could be achieved. The skin itself was not of interest in this particular study.

Specific comments:

Abstract

L23: Use: a "poor" correlation instead of "lower".

Response: Amended (replaced ‘lower’ with ‘poor’), line 23

L28: Avoid using the term "gold standard" for the MRI measurements in a general way (see general comments). Describe the comparison with MRI in a neutral way, based on facts (data of the MRI imaging system, image resolution etc.) and not based on this sloppily used term.

Response: The term "gold standard" has been removed, line 28 and throughout the manuscript.

Introduction:

L42: Use a technically correct and neutral description for the comparisons you made. At least for measurements of SF thickness, when performed correctly (which is not the case in this publication), US can be seen as the method with the highest accuracy; this should be discussed in detail using the relevant citations (in the Introduction and in the Discussion sections). The latest publications dealing with accuracy and reliability of US measurements for thickness measurements of SAT are missing, e.g. the recent papers published in the Br J Sports Med (Müller et al., 2017), or in Ultrasound in Med&Biol (Störchle et al., 2017).

Response: Text has been amended accordingly: 

"The latter represents the most appropriate standard currently available for testing the validity against other methods". line43-44

" Accuracy and reliability of US imaging for measuring anterior thigh SAT thickness have been recently demonstrated by Müller et al. [12], and Störchle et al. [13]. line 46-47

43: It is misleading to call the method that is the most expensive one automatically the "gold standard" (although, this misuse of this term "gold standard" is widely spread).

The MRI measurements made, at least for SAT measurements and with the MRI system used, do not represent "the standard" for testing validity, in the opposite, US – with its much better image resolution – is the standard for the purpose of SAT thickness measurements (it has been explained in the literature why US, when applied in standardised way, results in a thickness measurement accuracy (about 0.1 mm at 18 MHz) that is currently not reached by any other method including MRI (due to the much larger pixel/voxel sizes in MR images).  For example: the authors used a slice thickness of 7 mm, this value is almost 2 orders of magnitude above the US image resolution.

Response: The authors agree that  the term "gold standard" is used inappropriately, particularly in relation to the accuracy of measurements, as US reaches 0.1 mm compared to resolution of 3 Tesla MRI of 0.2x0.2x1.0 mm. The term "gold standard" has been removed throughout the manuscript, to avoid misleading the readers.

The text has been amended: " Ultrasound techniques have improved over time, leading to imaging muscle tissue with resolutions up to 0.1 mm [30], which is better than an image obtained by an high field MRI of 3 Tesla, which reaches a resolution of 0.2x0.2x1.0mm [31]" line 59-61

46:  "reliability". Also, discuss the recent papers on reliability of SAT measurement with US in detail as this is of essential concern for the study presented here. Compare to Br J Sports Med (A: Müller et al., 2017) and to B: Ultrasound in Med&Biol (Störchle et al., 2017). 

You cite papers on reliability, but do not discuss the accuracy obtainable with MRI (which depends on the measurement parameters, the pixel size, and the tissue segmentation program used). The obtainable accuracy and reliability will later in the manuscript have to be discussed in the context of the particular MRI setting the authors have used.

Response: As US is an operator-dependent procedure, ensuring intra and inter reliability of measurements is of great importance, and such studies are generally carried out before testing the validity of the procedure versus another methodology, and need to be established for specific muscles and tissues.

Of course the accuracy obtainable with MRI depends on the measurement parameters, the pixel size, and the tissue segmentation program used.

The intent of the present study was testing US imaging versus MRI for measuring thigh muscle, SAT, and fascia thickness and not particularly studying the accuracy of MRI, and so this aspect was not analyzed.

50: avoid  the term "gold standard", or give detailed explanations why you mean that this is the "gold standard" (in terms of accuracy and reliability at given MRI machine settings). Just writing that MRI is "seen as the gold standard" (by other authors) is not a scientific argument. Don't follow unreflected statements of other authors without analysing them on a scientific basis. 

Response: The term "gold standard" has been removed.

Materials and Methods:

2.3 US:

Specify the speed of sound you used for thickness measurements. Consider that speed of sound differs significantly from 1540 m/s (as used in most US machines) in fat (you cited ref. #37, but did not use the information contained in this publication concerning the analysis of accuracy; also see ref A).

Response: the speed of sound was 1540 m/sec

80: The authors obviously did not apply any pressure: according to Fig. 4, the authors used the method for avoiding compression artefacts as described in ref. 37 (a thick gel layer above the skin), but they do not cite this methodical procedure aspect as described in the ref. #37 of their manuscript (and in ref. A, too).

Response: Citations of  this methodical procedure has been amended in the text (line 89) and in References.

The authors should describe in detail how they measured the thicknesses in the US images (just one measurement with the cursor? average of how many measurements? What sound speed was used in the machine? What thickness measurement error is to be expected because of the wrong speed used, etc.)

Response: The US machine used in the present study (Esaote MyLab 25) assumes a constant propagation speed of 1540 m/sec and cannot be modified by the user.

It is indeed correct that propagation velocities are different in different tissues, and so speed displacement artifact occur when the sound wave passes through tissues of differing propagation speeds, and this represents an intrinsic limitation of the methodology, as in all studies of this nature, and it was not possible to factor in relative thickness measurement error in the study.

It has now been inserted in the text that US and MRI images were measured off-line twice and the mean of the two was used in the analysis. line 100-101

2.4 MRI:

Information on the field of view and on the pixel/voxel size and form is missing. Why did the authors not use the DIXON technique to show the skin separately? (maybe to be discussed in "Limitations").

Use precise terms: you measured SAT and skin thickness together – this is not identical to "subcutaneous fat (SF)". Use precise wording. Further, define clearly what you mean by "non-contractile tissue" when you use this term for the first time. Describe the core parameters of the image analysis software and what software setting was used in your study.  

93: wording again

Response: Text has been amended with the following information:

FoV 180*180, pixel spacing 0.3515625\0.3515625 line 97

- DIXON technique was not used to show skin separately, because it was not the intention to measure skin at all, as it is not relevant in the area of rehabilitation where this technique of measuring relative thickness of muscle and non-contractile tissue is of interest. Ultimately the aim is to assess muscle status and indicators affecting muscle function and mobility. The authors appreciate that in other fields, such as dermatology, it would be vital to measure skin separately and use the DIXON technique.

- SAT was measured without skin. (Measurements of tissues layers are now amended in the Figure 3 and 4, because it was not clear)

Definition of "non-contractile tissue" is now amended, when the term was used for the first time in the abstract, "(SAT combined with perimuscular fascia) line22-23

Results:

The results are interesting (although the methods were not used in a state-of-the-art manner) and statistical analyses appear to be correct and useful. The authors should point out that mean SAT thickness was larger when measured with US (and in the discussion, the authors should explain why this was the case – because of the wrong speed of sound for fat; compare to ref # 37 and ref A). It seems that the authors did not read their cited literature thoroughly; otherwise, they would have figured this out themselves already.

Figures:

Figs. 3 and 4: The depth of the image is not indicated.

Fig. 7: define what is meant by non-contractile tissue (remark: "SF" is also "non-contractile" !)

Description on axes is hard to read (letters too small).

Each figure with the figure legend should be self-explaining.

Response: The Figures have been amended accordingly

Discussion:

141: gold st. again

Response:  Removed

145: Here, I recognise the definition of non-contractile tissue for the first time. This should occur earlier.

Response:  Definition of "non-contractile tissue" is now amended, when the term was used for the first time in the abstract, "(SAT combined with perimuscular fascia) line22-23

159: Fasciae are complex structures, sometimes consisting of several layers, their thicknesses can change from one mm to another and so on (e.g. described in detail in: C Stecco: Functional Atlas of the Human Fascial System, 2015, Elsevier)  – this can be detected by US but not by MRI, at least not with the MRI machine and setting used. The authors should point out that MRI (their machine and their setting, and also what can be obtained with state-of-the-art MRI machines), therefore, cannot reach the accuracy of US (provided the appropriate speed of sound is used).

Response: The text has been amended as follows: "Another potential issue regarding the poor correlation and agreement results for fascia could be due to the limitation of the low field MRI machine used (0.35T). Also, while the resolution of the US image of the fascia was good,  the frequency was set at 7.5 MHz to improve ultrasound penetration and visualize deeper structures, at the expense of the image resolution of the superficial tissues where a higher frequency would have been optimal" lines 168-172.

170-191: The authors embed their results into other research findings in a suitable way and draw useful conclusions.

180: ".. to change in muscle and SF thickness with older age [35-39]."

Several of the cited papers do not deal with aging studies at all. What do the authors mean here? These citations are in a wrong context.

Response:  Citations have been amended

192: Limitations: The limitations of the authors' approach from the technical point of view are missing. It is also missing that skin and "SF" (this term should be replaced or clearly defined) are not separately treated (skin thickness depends substantially on the site used, and also on the person under study). It is also missing why the authors did not use a standardised measurement approach as was described for SAT measurements in the literature. It has to be noted that their thickness measurements start out from a speed of sound that is not matched with the given tissues – in this concern, the authors can argue that conventional machines are not able to choose the appropriate speed of sound and therefore their approach mirrors the usual medical B-mode US applications.   

Response: these comments have been dealt with in earlier points above. It is certainly true that the present technique mirrors the usual medical B-mode US applications and that was the intention.

Round 2

Reviewer 2 Report

The authors have improved all points that I have raised. The manuscript can be published now.